# Exploring Novel Therapeutic Targets in the Common Pathogenic Factors in Migraine and Neuropathic Pain

**DOI:** 10.3390/ijms24044114

**Published:** 2023-02-18

**Authors:** János Tajti, Délia Szok, Anett Csáti, Ágnes Szabó, Masaru Tanaka, László Vécsei

**Affiliations:** 1Department of Neurology, Albert Szent-Györgyi Medical School, University of Szeged, Semmelweis u. 6, H-6725 Szeged, Hungary; 2Doctoral School of Clinical Medicine, University of Szeged, Korányi fasor 6, H-6720 Szeged, Hungary; 3Danube Neuroscience Research Laboratory, ELKH-SZTE Neuroscience Research Group, Eötvös Loránd Research Network, University of Szeged (ELKH-SZTE), Tisza Lajos krt. 113, H-6725 Szeged, Hungary

**Keywords:** migraine, neuropathic pain, calcitonin gene-related peptide (CGRP), transient receptor potential (TRP) ion channels, endocannabinoids, glutamate, kynurenine, cytokines, neuroinflammation, microglia

## Abstract

Migraine and neuropathic pain (NP) are both painful, disabling, chronic conditions which exhibit some symptom similarities and are thus considered to share a common etiology. The calcitonin gene-related peptide (CGRP) has gained credit as a target for migraine management; nevertheless, the efficacy and the applicability of CGRP modifiers warrant the search for more effective therapeutic targets for pain management. This scoping review focuses on human studies of common pathogenic factors in migraine and NP, with reference to available preclinical evidence to explore potential novel therapeutic targets. CGRP inhibitors and monoclonal antibodies alleviate inflammation in the meninges; targeting transient receptor potential (TRP) ion channels may help prevent the release of nociceptive substances, and modifying the endocannabinoid system may open a path toward discovery of novel analgesics. There may exist a potential target in the tryptophan-kynurenine (KYN) metabolic system, which is closely linked to glutamate-induced hyperexcitability; alleviating neuroinflammation may complement a pain-relieving armamentarium, and modifying microglial excitation, which is observed in both conditions, may be a possible approach. Those are several potential analgesic targets which deserve to be explored in search of novel analgesics; however, much evidence remains missing. This review highlights the need for more studies on CGRP modifiers for subtypes, the discovery of TRP and endocannabinoid modulators, knowledge of the status of KYN metabolites, the consensus on cytokines and sampling, and biomarkers for microglial function, in search of innovative pain management methods for migraine and NP.

## 1. Introduction

Migraine and neuropathic pain (NP) are chronic pain syndromes with extensively studied pathogeneses. While their clinical manifestations strongly differ, their pathophysiologies have common roots. Cause-based treatment of these two devastating, painful neurological diseases is still unsatisfactory. Migraines are one of the frequent primary headache disorders with the typical clinical features of unilateral throbbing or pulsating and moderate to severe headaches with concomitant symptoms such as nausea, vomiting, photophobia, phonophobia, osmophobia, and allodynia. Its main subtypes are migraine without aura (M0) and with aura (MA). Episodic (EM) or chronic (CM) forms can be differentiated based on the number of migraine days per month [1]. NP is a chronic secondary pain condition caused by a lesion or disease in the central or peripheral somatosensory system [2,3,4]. It is characterized by burning and lancinating pain with an abnormal sensation, such as paresthesia, dysesthesia, or allodynia.

We underline the difference between these two painful conditions as follows: the quality of the pain (throbbing in migraine and burning in NP), the associated symptoms of migraine (nausea/vomiting, photophobia, and phonophobia) which are absent in NP, the action of triptans (very effective in migraine, ineffective in NP), and the effect of nonsteroidal anti-inflammatory drugs (effective in migraine, ineffective in pure NP). These two different clinical entities meet via their common pathomechanisms, such as hyperexcitability and sensitization, which involve neuropeptides (mainly the calcitonin gene-related peptide, CGRP), transient receptor potential (TRP) ion channel alterations, the endocannabinoid system, the glutamatergic system, pro- and anti-inflammatory cytokines, and microglia activation [5,6,7,8,9,10,11,12,13,14,15,16].

CGRP is a vasodilator neuropeptide that plays a crucial role in the pathomechanism of migraine. CGRP is well-documented in pain transmission in the somatosensory nervous system. The latest therapeutic innovation is based on human and fully humanized monoclonal antibodies (mAbs) targeting CGRP and CGRP receptors. Clinical trials in EM and CM patients revealed the high efficacy and safety of these pharmacons [17]. Clinical data has shown that anti-CGRP mAbs provide strongly efficacious preventive treatment for both EM and CM. For NP with co-existing CM, only one study showed a decrease in Neuropathic Pain Scale scores [18]. TRP channels are involved in pain mechanisms. Several clinical observations have indicated that different agents (e.g., herbs, food, environments) have the ability to influence migraine headaches via the modulation of subclasses of TRP superfamilies (TRP-ankyrin 1—TRPA-1; TRP-vanilloid 1—TRPV-1; TRP-melastatin 8—TRPM-8) [13,19]. Research targeting TRP has led to the innovation of the high-concentration (8%) capsaicin dermal patch for different types of NP, such as painful diabetic neuropathy (PDN), postherpetic neuralgia (PHN), and human immunodeficiency virus (HIV)-associated neuropathy [20].

Elements of the endocannabinoid system exert antinociceptive effects through the activation of cannabinoid receptors (CBR). Favorable preclinical results, which showed a decreased trafficking of pain transmission, were clinically confirmed in migraine patients. In the field of NP, only a few clinical studies are available [21]. The glutamate system is involved in the pain processes of hyperexcitability and sensitization. Kynurenines (KYNs) play pivotal roles in this process, since kynurenic acid (KYNA) is one of the rare endogenous antagonists of excitatory glutamatergic N-methyl-D-aspartate (NMDA) and α-amino-3-hydroxy-5-methyl-4-isoxazolepropionic acid (AMPA) receptors. The function of the KYN pathway in migraine and NP, based on the available human data and clinical trials, are summarized in this review [7,22].

The key pro-inflammatory cytokines are interleukin (IL)-1, IL-2, IL-6, IL-17, IL-18 (previously interferon gamma), and tumor necrosis factor alpha (TNF)-alpha, while the anti-inflammatory ones are IL-4, IL-10, and IL-37. There are controversial results with cytokines in the field of migraine, while in NP, both pro- and anti-inflammatory cytokines are of significant importance [15,23]. Microglia interact with the neuron. In migraineurs, altered levels of S100B, a sensitive marker of glial cell injury, have been demonstrated, which point to the role of microglia in hyperactivation of the trigeminovascular system. Microglia play a fundamental role in NP transmission and in the sensitization process in the nervous system [9,10,24]. There is evidence that cytokines have pathophysiological roles in pain genesis and transmission.

Preclinical research reveals valuable information on human diseases by employing in vitro and in vivo models [25,26,27,28]. Data collected in-laboratory has made significant contribution to understanding the pathomechanism of human diseases from molecular to organismal levels in search of therapeutic targets [29,30,31,32,33,34,35,36,37,38,39,40]. Here, we highlight clinical studies of common pathogenic factors in migraine and NP with reference to preclinical data to explore potential therapeutic targets and clarify the current missing data for it to be complemented in the near future, in search of innovative pain management in migraine and NP.

## 2. Calcitonin Gene-Related Peptide Function in Migraine and Neuropathic Pain

The first clinical data on the importance of the trigeminovascular system in the pathomechanism of migraine revealed that CGRP plasma concentration was elevated in the external jugular vein during a migraine attack (Table 1) [41]. Later, increased CGRP plasma levels were also observed in the cubital vein during the ictal period as compared with those during the interictal period [42]. Experimental work on human trigeminal ganglia demonstrated the distribution of CGRP and CGRP receptors [43,44].

The trigeminovascular system forms a bridge between the cerebral dura mater and the vasculature of the meninges, cortex, and of second-order nociceptive neurons of the trigeminocervical complex (TCC) [48,49,50]. One putative mechanism for the activation of the trigeminovascular system suggests that the peripheral branch of the pseudounipolar neurons of the trigeminal ganglion is triggered by cortical spreading depression, affecting different brain areas such as the frontal regions. [51,52,53]. Thus, CGRP released in the peripheral and central branches of trigeminal neurons lead to the vasodilation and neurogenic inflammation of the meninges and to the activation of the second-order sensory neurons of the TCC. The second-order neurons then activate the third-order neurons in the thalamus [54].

### 2.1. Migraine

The clinical sign of trigeminovascular activation and hyperexcitability is allodynia, which is pain due to an innoxious stimulus. Allodynia can have a cephalic or extracephalic localization [55]. A double-blind crossover clinical trial revealed that the intravenous infusion of alpha-CGRP versus placebo caused delayed migraine-like headaches in migraineurs [56]. This was the first clinical observation that clearly demonstrated that CGRP can induce migraine attacks, leading to the development of small molecule CGRP receptor antagonists, gepants. The second generation gepants—ubrogepant, rimegepant, and atogepant, which are orally administered—and the third generation version—vazogepant (not approved in the United States), which is intranasally applied —are available for the acute and/or prophylactic treatment of migraine [57,58,59].

A novel pharmacological innovation produced human and fully humanized mAbs against CGRP and CGRP receptors to prevent EM and CM. Eptinezumab, fremanezumab, and galcanezumab selectively bind to the CGRP itself as a ligand, while erenumab competitively and reversibly targets CGRP receptor components. All of the above-mentioned mAbs are highly effective and safe in the prophylaxis of both EM and CM [5,17,60,61,62,63].

### 2.2. Neuropathic Pain

Human studies have found that nerve fibers in the spinal cord laminas I, III, and V had CGRP-like immunoreactivity, and that their receptors are widely distributed in the pain pathways of the nervous system [64,65]. Experimental data has shown that CGRP can sensitize nociceptors and also can induce central sensitization [65]. Allodynia is a clinical feature of central sensitization and is one of the most common sensory abnormalities indicating NP. In spite of these preclinical findings, the available clinical data in this area are very sparse.

In skin biopsy samples of PHN patients, increased CGRP levels were found in keratinocytes [47] (Table 1). Complex regional pain syndrome (CRPS) occurs in two types, both occurring after trauma, and peripheral nerve injury exists only in CRPS type 2. The latest classification by the International Association for the Study of Pain (IASP) suggests that CRPS type 2 may be associated with neuropathic mechanisms [66]. Clinical studies have demonstrated increased CGRP serum levels in patients suffering from CRPS [46,67]. In patients with painful neuroma, higher densities of CGRP-immunoreactive nerve fibers were observed in comparison to controls [45,68]. A retrospective clinical study validated the effectiveness of CGRP-targeting mAbs in CM patients who also suffered from NP. Interestingly, in these patients, the anti-CGRP treatment significantly improved the Neuropathic Pain Scale scores. Limitations of this study include the fact that it was an open-label trial and not placebo-controlled trial, and that the number of patients was very low [18]. Allodynia is the common clinical feature of both migraine and NP. Peripheral and central sensitization, allodynia, and responsiveness to anti-CGRP mAbs point to the potential common role of CGRP in both migraine and NP.

## 3. Transient Receptor Potential Ion Channel Function in Migraine and Neuropathic Pain

TRP ion channels are non-selective cation channels and can be divided into six subfamilies: TRPV (vanilloid), TRPA (ankyrin), TRPM (melastatin), TRPC (canonical), TRPP (polycystin), and TRPML (mucolipin) [69,70]. Preclinical studies have concluded the putative role of TRPs in migraine. The activation of TRPs (TRPA-1, TRPV-1) in the TCC results in CGRP and substance P release and depletion from the central branch of the trigeminal nerve endings. The consequences are overexcited, second-order, pain-processing neurons in the TCC, which lead to central sensitization. The peripheral parts of the trigeminal nerve terminals, projecting into the cerebral dura mater and the vasculature of the meninges, contain TRPs. The activation of the TRPs leads to CGRP release, which can act on its receptors on the smooth muscle cells of blood vessels, resulting in strong vasodilation [70,71].

### 3.1. Migraine

Capsaicin, as a potent and highly selective TRPV-1 receptor agonist, is a chemical compound isolated from chili pepper. A randomized controlled trial (RCT) using intranasal civamide (a synthetic stereoisomer of capsaicin) for M0 and MA patients during headache attacks revealed decreased pain severity at 2 h post-dose in 55.6% of patients, and 22.2% of patients were pain-free [72]. A double-blind study of CM patients demonstrated that repeated intranasal capsaicin application had favorable effects [73]. A single-blind, placebo-controlled, crossover study of a small number of M0 patients, who were treated with topical capsaicin (0.1%) jelly, led to the relief of arterial pain by at least 50% [74].

TRPM-8, a non-selective cation channel, can be activated by cold temperatures and menthol. A triple-blind RCT revealed that a 10% menthol solution applied to the forehead and the temporal skin areas was significantly superior to the placebo at providing 2 h long pain freedom [75].

Extensive preclinical studies focusing on TRP ion channels have concluded that TRPA-1 and TRPV-1 could play crucial roles in the activation of several substances (as migraine triggers) that evoke migraine pain. Odors (cigarette smoke, formalin, *Umbellularia californica*—‘headache tree’) are can trigger and worsen migraine attacks via TRPA-1 receptor agonism. Other agents, such as *Tanacetum parthenium* (feverfew) and *Angelica sinensis* (dong quai, female ginseng) as desensitizing TRPA-1 receptor agonists, are migraine-preemptive factors. A well-known migraine trigger, glyceryl trinitrate (nitroglycerine-NTG) as a nitric oxide donor, is also a TRPA-1 receptor agonist [19]. The long-recognized migraine triggers are alcohol-containing drinks (ethanol), which are TRPV-1 receptor agonists. Capsaicin, as a pungent ingredient of paprika (*Capsicum*), is a desensitizing agonist of the TRPV-1 receptor [19].

An unusual clinical study examining the scalp arteries (superficial temporal and occipital arteries) of CM patients demonstrated significantly increased TRPV-1-like immunoreactive nerve fiber density in the wall of the arteries of CM patients versus in those of the control group(Table 2) [76].

A recent pilot study searching for predictors of migraine chronification investigated 1911A/G single nucleotide polymorphism (SNP) in the TRPV-1 gene in patients with EM and CM compared to healthy subjects. The results showed that genotype frequency distribution in EM was comparable with that in the controls, while it differed significantly in CM patients [80]. Another genetic study on Spanish migraine patients demonstrated an association between migraine and SNPs of the TRPV-1 and TRPV-3 receptor genes [13,81]. The above-discussed clinical data has led to the design of early-phase clinical trials targeting thermo TRP channels for migraine treatment, such as an oral TRPV-1 receptor antagonist (NCT00269022), a TRPM-8 receptor agonist (topical menthol 6%) (NCT01687101), and a TRPM-8 receptor antagonist (oral AMG 333) (NCT01953341) [82]. The final results are not yet available.

### 3.2. Neuropathic Pain

Nociceptors are special afferent sensory neurons which convey thermal, mechanical, and chemical stimuli. The members of the TRP family are densely expressed on nociceptors. Therefore, they have fundamental roles in nociception and NP transmission [83,84,85,86]. A high-concentration (8%) capsaicin patch reversibly de-functionalizes the nociceptive nerve terminals [14,87]. Based on the latest Cochrane Database, the 8% capsaicin patch is effective, well-tolerated, and safe for the treatment of PHN, HIV-neuropathy, and PDN (Table 2) [20]. Clinical trials revealed that the 8% capsaicin patch significantly reduced the average pain intensity in chronic postsurgical NP [77]. A retrospective observational study collecting different types of peripheral NP patients (PHN, chronic postsurgical NP, post-traumatic NP, PDN, HIV-associated NP, painful radiculopathy, and trigeminal neuralgia) demonstrated a reduction in pain intensity and in the pain area after the application of the 8% capsaicin patch [78]. The 8% capsaicin patch can provide pain relief for up to 3 months or longer after a single 30–60 min application in chemotherapy-induced NP [79].

Early phase clinical studies targeting thermo TRP channels for NP treatment, including an intranasal TRPV-1 receptor agonist in PHN (NCT01886313), a subcutaneous TRPV-1 receptor inhibitor (parentide) in non-specified NPs (EP002846-21), an oral TRPV-3 receptor antagonist for non-specified NPs (NCT01463397), and a TRPM-8 receptor agonist (topical menthol 7%) for chemotherapy-induced peripheral neuropathy (NCT0185567) are under way [82]. Published data are not yet available.

Capsaicin, as a potent TRPV-1 receptor agonist, can decrease the intensity of pain either in migraine or NP via the modulation of the release of pain-related neuropeptides from nociceptors. In migraine, several agents targeting TRPA-1 and TRPV-1 receptors can trigger or preempt headache attacks. The high-concentration (8%) capsaicin patch is strongly recommended for the treatment of peripheral NPs such as PDN, PHN, and HIV-neuropathy [88,89]. Early-phase clinical trials are ongoing both for migraine and NPs.

## 4. Endocannabinoid Function in Migraine and Neuropathic Pain

Endocannabinoids are endogenous cannabis-like substances. Chemically, they are characterized as small molecules, and they are derived from arachidonic acid. As neurotransmitters, endocannabinoids are part of the biological endocannabinoid system and act on CBRs: CBR type 1 (CBR-1) and CBR type 2 (CBR-2). Their main endogenous ligands are anandamide (N-Arachidonoylethanolamine) and 2-arachidonoylglycerol (2-AG). Moreover, the system involves enzymes that regulate the synthesis and degradation of the ligands. CBR-1 is located in the nervous system, mainly in the brain, while CBR-2 is found in the immune system [20,90].

In the endocannabinoid system, one of the main catabolic enzymes is fatty acid amide hydrolase (FAAH), which catabolizes fatty acid ethanolamides such as anandamide. Other enzymes of this system include monoacylglycerol lipase (MAGL), diacylglycerol lipase alpha, diacylglycerol lipase beta, and alpha/beta hydrolase domain 6. Fatty acid ethanolamides and 2-AG are the main endocannabinoid signaling lipids interacting with CBR-1 and CBR-2 [20,91]. The endocannabinoid system seems to be dysfunctional and dysregulated in migraine. It interacts with migraine-related pathways such as the serotonin system (5-HT_1A_, 5-HT_2A_ receptors), the modulator of somatic pain transmission (periaqueductal grey matter), meningeal vessel dilatation, and the activation of the TCC [92,93].

### 4.1. Migraine

In a clinical trial, the activity of FAAH and the specific anandamide membrane transporter (AMT) were measured in platelets taken from the peripheral blood of M0 patients and healthy controls. The results showed significant sex differences in the activity of FAAH and AMT in both study groups. Namely, an increase in the activity of FAAH and AMT was found only in female but not male M0 patients (Table 3) [94]. A study focused on the examination of anandamide, palmitoylethanolamide (PEA), and 2-AG concentrations in the cerebrospinal fluid (CSF) of patients with CM compared to those in the CSF of control subjects. The results showed that CSF concentrations of anandamide were significantly lower, while those of PEA were significantly higher in CM patients versus in the non-migraineur control group. 2-AG concentrations were below detection level in both patient and control groups [95].

A comparative clinical trial demonstrated that levels of AMT and FAAH were significantly reduced in the platelets of CM patients compared to those in the platelets of EM patients and the control group, and this was observed for both sexes [96]. A clinical study investigating the levels of anandamide, 2-AG, and serotonin in the platelets of CM patients and healthy controls found that anandamide and 2-AG platelet levels were significantly lower in CM patients versus controls. Furthermore, serotonin levels in the platelets were also strongly reduced in the CM group and were correlated with 2-AG levels [97]. An observational, cross-sectional study comparing the binding of CBR-1, as detected by positron emission tomography (PET), among female migraine patients and healthy controls demonstrated a global increase, which was most pronounced in the anterior cingulate, mesial temporal, prefrontal, and superior frontal cortices of the brains of migraineurs [100].

A clinical trial was designed for migraineurs with medication overuse headaches before and after withdrawal treatment. The results demonstrated a marked facilitation of spinal cord pain processing (an increased temporal summation threshold of the nociceptive withdrawal reflex and a reduction in the related pain sensation) in migraineurs before withdrawal treatment when compared with controls. The significant acute reduction of FAAH activity in platelets was coupled with a reduction in the facilitation of pain processing after versus before withdrawal treatment [101]. A clinical study concluded that the plasma levels of anandamide and related N-acylethanolamines and linoleic acid-derived oxylipins did not show any differences between M0 versus MA patients and migraine versus healthy controls [98].

Results from a genetic study demonstrated a significant haplotypic effect of *CNR1* (the gene of CBR-1) on headaches with migraine symptoms (e.g., nausea, photophobia, disability) only when using extreme trait combinations (0 symptoms versus 3 symptoms) [102]. Later, the same research group reported that variants in the *CNR1* gene were predisposed to headaches with nausea in the presence of life stress. None of the SNPs showed the primary genetic effects on possible migraine [103]. A RCT testing the effects of a 12-week aerobic exercise plan on plasma anandamide concentration and its relationship with clinical and cardiorespiratory outcomes in EM patients revealed plasma anandamide level reduction both in migraine and control exercise groups. Significant correlations were found between the reduction in abortive medication used and in cardiorespiratory fitness and reduced anandamide plasma levels [104].

A recent pilot study of EM and CM with medication overuse headache patients demonstrated higher CBR-1 and CBR-2 gene and protein expression in peripheral blood mononuclear cells compared to controls. FAAH gene expression was lower in the migraine groups compared to that in the controls. The gene expression of MAGL was significantly higher in the migraineurs [105]. In a clinical trial of EM patients and healthy controls, plasma anandamide and PEA levels, the latter being an anandamide activity enhancer, and spinal sensitization were evaluated in a validated human model of migraine based on systemic NTG administration. After NTG administration, anandamide plasma levels were increased in both groups, while increased PEA plasma levels were detected only in the EM group [106].

### 4.2. Neuropathic Pain

Endocannabinoids exert effects on a wide range of biological cell functions, such as exocytosis, proliferation, differentiation, and the control of pain transmission via inhibiting the ascending stimulatory pain pathways and activating the descending inhibitory pain pathways [107,108]. An early clinical study of patients suffering painful carpal tunnel syndrome demonstrated that treatment with PEA (600 mg or 1200 mg administered daily for 30 days) significantly reduced the median nerve latency time during nerve conduction tests [109]. In a later RCT, ultramicronized PEA treatment (administered sublingually) was investigated in patients with spinal-cord-injury-associated NP. The results showed no difference in mean pain intensity between ultramicronized PEA and the placebo treatment (Table 3) [100]. Based on promising preclinical data on FAAH and MAGL, clinical trials on MAGL inhibitors are ongoing. A randomized, placebo-controlled, optimized titration study with a MAGL inhibitor (ABX-1431) in PDN patients (NCT03447756) and a double-blind, placebo-controlled, crossover trial in central (multiple sclerosis-associated) NP patients (NCT03138421) are being conducted. In both clinical trials, favorable safety profiles were observed. Detailed results related to efficacy are not yet available [20]. The importance of the endocannabinoid system in pain modulation has been known since the early 1990s. Based on clinical data on the ligands and enzymes in this system, a correlation has been shown between the endocannabinoid system and migraine. Only limited clinical data on how the compounds of this system affect NP are available. In the future, CBR antagonists and FAAH and MAGL enzyme inhibitors might be promising therapeutic targets in the treatment of both migraine and NP.

## 5. Kynurenine Function in Migraine and Neuropathic Pain

The KYN pathway is the metabolic pathway of tryptophan (Trp) catabolism. The determinative Trp degradation product is L-kynurenine (L-KYN), which serves as a precursor for KYNA. KYNA is one of the rare endogenous antagonists of excitatory amino acid receptors. By affecting glutamate receptors, it has a role in pain processing and neurogenic inflammation [6,7,8,110,111], as well as in cognitive dysfunctions [112,113,114,115,116]. The sites of central sensitization are the second-order neurons of the TCC. This sensitization is induced by the release of glutamate from C-fibers of the central branch of pseudounipolar trigeminal neurons. Calcium ion influx and opened calcium storage in the cells result in increased intracellular calcium ion levels, which activate protein kinase C and lead to the phosphorylation of NMDA receptors. This process results in increased glutamate sensitivity, which is the basis for the hyperexcitability of the neurons [6,117].

### 5.1. Migraine

Related to the above-mentioned process, clinical studies were performed using different body fluids including plasma, serum, CSF, and saliva. Higher plasma glutamic acid levels were observed both during attacks and pain-free periods in M0 and MA patients (Table 4) [118]. High levels of glutamic acid in platelets were detected in patients with MA compared to M0 patients and healthy controls. Furthermore, glutamic acid platelet concentrations were higher ictally in MA patients [119].

In EM M0 and MA patients, plasma levels of glutamic acid were lower during attacks, while CSF concentrations of glutamic acid were higher in the migraineurs than in the controls [120]. Interictally, in the saliva of M0 patients, elevated glutamic acid concentrations were reported [121]. High glutamate concentrations in the CSF of CM patients compared to controls were also demonstrated [122].

Imaging studies in migraine patients have attempted to find a link between the glutamatergic system and specific brain regions. Altered excitatory neurotransmitter distribution in the anterior cingulate cortex and insula of migraineurs was observed by magnetic resonance imaging spectroscopy (MRI) [127]. A meta-analysis of MRI spectroscopy data during pain-free periods in patients suffering from M0, MA, or CM revealed increased glutamate concentrations in particular brain regions [128,129]. In M0 patients, during a resting state functional MRI, altered periaqueductal gray matter functional connectivity (as a brainstem migraine generator and a pain modulator) was detected and found to be correlated with plasma Trp concentrations, both of which were higher in migraineurs than controls [130,131,132].

A rare subtype of MA, familial hemiplegic migraine (FHM), can be divided into three subclasses: FHM1, FHM2, and FHM3. The following genetic mutations led to the alteration of the glutamate system: In the patients suffering from FHM1, *CACNA1A* (encoding the α1 subunit of the neuronal Ca,2.1. voltage-gated calcium channel) gene disruption results in glutamate release from presynaptic nerve terminals. In FHM2 patients, the *ATP1A2* gene (encoding the α2 subunit of Na^+^/K^+^ adenosine triphosphate (ATP)-ase pumps) is damaged and indirectly reduces the uptake of glutamate from the synaptic cleft in astrocytes. In FHM3, the *SCNA1* (encoding the pore-forming α1 subunit of neuronal Na_V_1.1 Na^+^ channels) gene lesion can reduce the firing of inhibitory interneurons and can increase glutamate levels in the synaptic cleft [133,134,135,136]. By studying these rare subtypes of MA, the role of glutamate has become better characterized.

CM is a distinct subclass of migraine that develops if the patient suffers from more than 15 headache days per month, which is accompanied by at least eight days of M0 or MA for three consecutive months [1]. In CM, which is a devastating form of migraine headache that greatly affects quality of life, altered KYN pathway metabolites and a reduction in the serum levels of KYNA have been observed [137].

In a well-designed clinical trial examining female M0 patients during headache-free periods, plasma concentrations of Trp metabolites (L-KYN, KYNA, anthranilic acid, picolinic acid, and 5-hydroxy-indoleacetic acid) were significantly decreased. Diminished peripheral Trp catabolism during the interictal period might act as a trigger of migraine attacks [123]. The first in-human, phase 1, open-label, single ascending dose study of L-KYN administered via intravenous infusion in healthy volunteers revealed that L-KYN was safe and well-tolerated [124]. Thus, monitoring the status of KYN metabolism is under extensive research [138,139].

### 5.2. Neuropathic Pain

A clinical sign of central sensitization is the phenomenon of allodynia, which mirrors the activation of the glutamatergic system in NP [6,22,140]. Overactive glutamatergic transmission via NMDA receptors is the basis of central sensitization in NP. Blocking the allosteric glycine B co-agonist site on NMDA receptors leads to the antagonism of the glutamate system. L-4-chlorokynurenine, a novel oral prodrug, is a potent and selective glycine B site inhibitor.

A crossover RCT revealed that NGX426, an oral AMPA-kainate receptor antagonist, reduced capsaicin-induced pain and hyperalgesia in healthy volunteers [141]. A phase 2 outpatient RCT examining LY545694 tosylate, a potent and selective ionotropic glutamate receptor antagonist, in PDN patients did not demonstrate a difference when compared to the placebo [142]. A dose-escalation RCT demonstrated a consistent reduction of allodynia and mechanical and heat hyperalgesia in an intradermally capsaicin-induced pain model in healthy volunteers [143].

In CRPS patients, the plasma levels of L-glutamate significantly increased, whereas those of L-trp significantly decreased when compared to the controls. The L-KYN to L-trp (KYN/Trp) ratio exhibited a significant increase in patients. A significant correlation between overall pain, plasma levels of L-glutamate, and the KYN/Trp ratio was detected. A correlation between the decrease in plasma L-Trp concentration and the disease duration was also observed in CRPS patients (Table 4) [125].

An exploratory pilot study involving female patients with NP-like syndromes, such as temporomandibular disorders myalgia and fibromyalgia, showed associations between the KYN/Trp ratio and pain intensity. In the temporomandibular disorder myalgia, a significant negative correlation between plasma Trp concentration and the worst pain intensity was observed, and a positive correlation between the KYN/Trp ratio and both the worst and average pain intensities were observed. Women suffering from fibromyalgia exhibited significantly lower plasma Trp levels than the controls did [126].

In addition to neuropeptides, the pathomechanism of hyperexcitability and sensitization is an overactivated glutamate system both in migraine and NP. Alteration of the KYN system has been reported in these two painful clinical conditions. Metabolites of the KYN pathway might have future therapeutic potential for migraine and NP.

## 6. Cytokine Function in Migraine and Neuropathic Pain

In the late eighties and early nineties, clinical studies demonstrated that intravenously administered TNF produced headaches in patients with tumors [144,145,146,147]. There is growing evidence that cytokines play a role in the genesis of migraine pain. They are released by neurons, microglia, astrocytes, macrophages, mast cells, and T-cells. Human studies reflect that the pro-inflammatory cytokines are TNF-alpha, IL-1beta, IL-6, and IL-18 [15,148]. Given that a balance of pro- and anti-inflammatory cytokines is important for neural functions [149,150,151,152], alterations in pro- and anti-inflammatory cytokines could be involved in synaptic and behavioral changes [153,154,155,156,157].

### 6.1. Migraine

The pro-inflammatory cytokines may have a role in inducing nausea and headaches during a migraine attack by increasing arachnoid acid production [15]. Several trials investigated the pro- and anti-inflammatory cytokines in plasma, saliva, and CSF of migraineurs. In an early clinical study, there were no differences in plasma IL-1 and TNF during migraine attacks compared to headache-free periods in M0 and MA patients (Table 5) [158].

A clinical trial analyzed IL-1beta, IL-6, and TNF-alpha in 24 h urine samples of female migraineurs during menstrual and non-menstrual migraine attacks and headache-free periods and compared them with those of non-headache controls. The mean IL-6 levels in the urine were higher in all three samples for migraineurs versus controls, while the IL-1beta levels showed no difference. The TNF-alpha values were very low in the menstrual migraineurs compared to those in the controls [158].

Another clinical trial failed to demonstrate differences in the serum concentrations of TNF-alpha and IL-6 between patients with M0, patients with MA, and healthy controls; however, the soluble receptor TNF-RI tended to be lower [160]. A clinical study of migraine patients revealed that the plasma levels of TNF-alpha, IL-1beta, and IL-10 were significantly higher ictally versus interictally [178]. The TNF-alpha levels in the internal jugular blood of M0 patients were elevated during ictal periods [161]. A clinical trial demonstrated that IL-10 serum levels were higher during migraine attacks versus during the interictal period and in healthy controls. Furthermore, the IL-6 serum concentrations were increased in the migraineurs compared to those in the controls [162]. A pilot study of EM patients demonstrated no significant difference in the serum levels of TNF-alpha during the attacks or headache-free periods [163]. A case-control study investigating newly diagnosed migraine patients revealed significantly higher serum IL-1beta and IL-6 concentrations, while the IL-10 serum levels were lower compared to those of healthy controls [164]. A clinical study investigating M0 and MA patients in both attack and pain-free periods revealed that serum levels of IL-6 were significantly higher in migraine patients during attacks compared to those in controls [165].

A prospective, case-control RCT of migraineurs concluded that the serum concentrations of TNF-alpha, IL-1beta, and IL-6 were significantly higher during migraine attacks compared to those in controls [166]. In the MOXY study, which studied female migraineurs responsive to adjunctive cervical non-invasive vagus nerve stimulation (VNS), the interictal saliva ELISA assays of IL-1beta showed significantly elevated values both pre- and post-VNS procedure when compared to healthy controls [167]. The evaluation of the inflammatory state in migraineurs versus healthy controls in a case-control study demonstrated that the serum levels of TNF-alpha and IL-6 were significantly increased in CM patients as opposed to EM patients and controls [168]. The SalHead longitudinal prospective cohort study, analyzing salivatory IL-6 and IL-1beta levels, observed non-significant differences at various time-points (headache-free period versus during a headache versus one day after the headache) between migraine patients and tension-type headache patients. Salivatory levels of IL-1beta had the highest discriminatory value between headache patients and healthy controls [169]. An investigation of serum IL-18 (previously interferon-gamma) levels in M0 and MA patients, ictally and interictally, revealed that they were higher in migraineurs than in the control group. IL-18 serum concentrations were similar in the ictal and interictal periods [170].

An interesting study analyzing pro- and anti-inflammatory cytokine levels in the CSF revealed that IL-1 receptor antagonist levels were elevated in M0 and MA patients during attacks compared to those in controls. There were significant differences in the CSF levels of certain cytokines (IL-1 receptor antagonist, monocyte chemoattractant protein-1, and transforming growth factor-beta1) between the migraine and episodic tension-type headache patients and the pain-free controls. The intrathecal pro-inflammatory monocyte chemoattractant protein-1 level was correlated with the IL-10 anti-inflammatory cytokine in MA patients [171].

The analysis of genetic variations of cytokines has provided useful data regarding the neuroinflammation process of migraine. A genetic study showed significant differences in the TNF-alpha −308G/A and IL-1beta +3953C/T gene polymorphisms in migraineurs versus control subjects [172]. A meta-analysis from 2011 focusing on TNF-alpha 308G/A and TNF-beta 252A/G gene polymorphisms among migraine patients concluded that there was no overall association between the above-mentioned gene variants and migraine [173]. Another meta-analyis, published in 2014, revealed that TNF-beta 252A/G gene polymorphism was not associated with overall migraine risk [174]. A clinical study analyzing omega-3 fatty acids and nano-curcumin supplementation targeting TNF-alpha gene expression and serum concentrations in migraine patients demonstrated that the TNF-alpha messenger ribonucleic acid (mRNA) was significantly downregulated and that the serum level of TNF-alpha was decreased [175].

A RCT of EM patients revealed downregulated IL-6 mRNA and decreased IL-6 serum concentrations [176]. An investigation of the IL-6 coding gene in the peripheral blood of M0 and MA patients demonstrated no significant differences in the expression of IL-6 between total migraine patients and healthy controls. However, the expression of IL-6 was significantly higher in the MA patients versus the controls [177]. A clinical study investigating cytokine-coding gene expression in blood among M0 and MA patients revealed that the expression of IL-4, tumor growth factor-beta (TGF-beta), and TNF-alpha was increased in patients compared to that in controls, but there was no difference in the expression levels of IL-1beta, IL-17, and IL-2. The expression of IL-18 was also higher in the migraineurs (lower in women than in men) compared to that in the healthy controls [178]. A genetic study focusing on the TNF-alpha gene polymorphisms (rs1800629 and rs1799724) among Jordanian migraineurs showed its significant associations with migraine occurrence [179]. For future therapeutic innovations in migraine, IL-37, as an anti-inflammatory cytokine, may be a crucial player. IL-37, as a natural inhibitor of immune response and inflammation, can diminish pro-inflammatory IL-1 activation and upregulate the anti-inflammatory IL-10 [15]. A recent meta-analysis of peripheral inflammatory cytokines in migraine concluded that IL-1beta, IL-6, and TNF-alpha serum levels were higher in migraineurs when compared to healthy controls, while IL-2 and IL-10 (an anti-inflammatory cytokine) did not show significant differences [180].

### 6.2. Neuropathic Pain

There is increasing evidence that cytokine expression is a contributor to NP [181]. In the development of NP, TNF-alpha, IL-1, and IL-6 may have fundamental roles in inflammation [129]. Cytokine action sites involve peripheral nerve endings, dorsal root ganglia, the synaptic junction in the dorsal horn of the spinal cord, and distinct regions of the brain (like the hippocampus, locus coeruleus, and red nucleus) [182,183,184,185].

A clinical investigation of nerve biopsies in neuropathic patients with and without pain revealed upregulated TNF-alpha expression in human Schwann cells in the group with pain (Table 4) [186]. A clinical study focusing on mRNA expression and the plasma protein levels of cytokines in patients who had painful versus painless neuropathies demonstrated that both of the measured parameters of pro-inflammatory cytokines (IL-2 and TNF-alpha) were increased in the patient group with pain, while the levels of IL-4 and IL-10, as anti-inflammatory cytokines, were lower in this group compared to that in the patients without pain [187]. A clinical investigation of PDN and diabetic neuropathic patients without pain indicated increased TNF-alpha serum levels in the neuropathic group compared to non-neuropathic and healthy groups [188].

A prospective genetic study analyzing local (skin) and systemic (plasma) cytokine gene expression in patients suffering from small fiber sensory neuropathy revealed that the local gene expressions of IL-6 and IL-8 (chemokine) were significantly increased (5-fold), while an only mildly elevated gene expression of IL-2 and IL-10 was detected in the plasma (2-fold) [181].

A cross-sectional study revealed that plasma TNF-alpha levels and immunoreactivity for TNF-alpha were higher in patients with severe pain, based on VAS in PDN patients, compared with controls [189]. A prospective RCT of patients suffering low back and leg pain, caused by lumbar disc herniation and lumbar spinal canal stenosis, who were treated with epidurally administered etanercept (an anti-TNF mAb) versus dexamethasone, demonstrated that etanercept significantly decreased both leg and low back pain [190]. The same clinician group published the results of a clinical trial using epidurally applied tocilizumab onto the spinal nerve as an anti-IL-6 receptor antibody for patients with low back and radicular leg pain caused by lumbar spinal stenosis. They concluded that the infiltration of tocilizumab was more effective than that of dexamethasone in these patient groups [191]. A double-blind, placebo-controlled trial evaluating the analgesic effect of losmapimod (a p38 alpha/beta inhibitor) in patients with NP after peripheral nerve injury revealed that losmapimod statistically did not differ in analgesic response to the placebo [192].

A prospective study of patients with painful or painless peripheral neuropathy demonstrated that painful neuropathies are associated with increased pro-inflammatory IL-6 and anti-inflammatory IL-10 gene expression in the sural nerve [193].

A clinical trial including patients with painful distal sensorimotor polyneuropathy (DSPN) from the German KORA F4 survey found positive associations between serum concentrations of IL-6 and painful DSPN, whereas no associations were observed with IL-18, TNF-alpha, and IL-1 receptor antagonists [194]. A parallel-group RCT of patients suffering central NP associated with spinal cord injury revealed that, in the anti-inflammatory diet treatment group, the serum levels of pro-inflammatory cytokines, such as interferon-gamma (later named IL-18), IL-1beta, and IL-6, were decreased [195]. A cross-sectional study assessing different serum biomarkers including cytokines (oncostatin, TNFSF10, TNFSG12, and TNFSF14) in patients with diabetic polyneuropathy did not find differences in biomarker levels between DSPN patients with and without pain [196].

A genetic trial focused on patients with and without NP after peripheral nerve lesioning revealed that, in white blood cells, the gene expression of TNF-alpha was higher in patients with pain compared to those without pain. IL-1beta gene expression was higher in the patients with pain compared to the controls. IL-10 showed lower gene expression in the group with pain than in the control group, and IL-4 gene expression was not different between the control and painless patients [16].

A pilot RCT of patients with peripheral NP due to PHN examined the mRNA expression of IL-6 in two study groups. In Group 1, patients with PHN-related NP received pregabalin monotherapy alone, while Group 2 patients were treated with a combination of pregabalin and cognitive behavioral therapy. The results showed that the patients in Group 2 had a significantly downregulated IL-6 mRNA expression compared to Group 1 [197].

A recent meta-analysis focused on the association between pro-inflammatory (TNF-alpha, IL-2, IL-6, IL-18) and anti-inflammatory (IL-10) cytokines as systemic inflammatory biomarkers in painful and painless diabetic neuropathy. It concluded that the serum levels of pro-inflammatory markers were increased, while those of the anti-inflammatory ones were lower in painful compared to painless diabetic polyneuropathy [23].

In migraine, whether in ictal or interictal phases, the data regarding cytokines are inconsistent, but pro-inflammatory cytokines tend to be elevated in human clinical trials. InNP patients, the levels of pro-inflammatory cytokines have also been shown to be elevated compared to those of controls in the majority of clinical studies.

## 7. Glial Function in Migraine and Neuropathic Pain

The trigeminovascular system is the backbone of the most accepted hypothesis for migraine pathogenesis. The center of this system is the trigeminal ganglion, which involves pseudounipolar neurons and satellite glial cells. There are strong data that glial cells have a role in peripheral sensitization and neuroinflammation, which lead to migraine chronification and the development of autonomic symptoms during migraine attacks [150,198].

### 7.1. Migraine

S100B is a calcium-binding protein in the cytoplasm of glial cells in the nervous system, and it is a sensitive marker for glial cell injury. Clinical studies have used it as a biomarker for the detection of glial involvement in the pathomechanism of EM and CM patients, ictally and interictally. Unfortunately, the results are inconsistent. In a clinical study which was performed during and after migraine attacks (2–4 days), the serum concentration of S100B was elevated (Table 6) [199]. A trial of M0 patients revealed increased serum S100B levels in both ictal and interictal phases [24]. A cross-sectional prospective study of M0 and MA patients revealed that serum S100B levels were significantly lower than those of controls, and the two study groups did not differ [200]. A pilot RCT of CM patients revealed that a glial cell modulator, ibudilast (phosphodiesterase inhibitor), did not reduce the frequency of headaches but was well tolerated [201]. In a case-control trial, serum levels of S100B were analyzed in EM and CM patients, and the results showed no interictal S100B elevation [202]. In the EM and CM patients, the serum level of S100B was significantly higher compared to that of the controls, and there was no difference between the two patient groups [203].

### 7.2. Neuropathic Pain

In the case of peripheral nerve lesions (the peripheral arm of neurons of the dorsal root ganglia), one of the main consequences is ATP release from the central terminals in the dorsal horn of the spinal cord. ATP acts on the microglia via purinergic P2 × 4 receptors and results in the release of brain-derived neurotrophic factor from the activated glial cells. This trophic factor stimulates second-order neurons via the activation of tyrosine kinase B receptors. The result of this process is the central sensitization of the second-order neurons, leading to the development of allodynia as a main clinical sensory sign of NP. It also leads to the overactivation of the third-order neurons in the thalamus [6,205,206,207,208,209,210]. Based on these findings, NP can be considered a gliopathy [211,212].

An early PET study, using a sensitive in vivo marker of glial cell activation, demonstrated activated glial cells in the contralateral thalamus after limb amputation, which pointed to a long-term transsynaptic glial response in the central nervous system (CNS) following peripheral nerve injury (Table 6) [204]. A functional imaging technology using newly synthesized glia-PET tracers has emphasized the importance of neuron-microglia interactions in the mechanism of NP [9].

Preclinical studies have confirmed that opioids could activate the microglia via the toll-like receptor 4 and the myeloid differentiation factor 2 receptor complex. The consequence was an activated mitogen-activated protein kinase (MAPK) system, which resulted in interleukin gene activation, leading to neuroinflammation [213,214].

Motor cortex stimulation is a potential therapeutic method for the relief of NP. In a clinical study, epidural strips were implanted over the motor cortex in central post-stroke NP patients and one trigeminal nerve injury NP patient. A comparison of postoperative PET with preoperative scans demonstrated significant decreases in a tracer, [(11)C] diprenorphine, binding to opioid receptors in different brain areas. Binding changes in the anterior middle cingulate cortex and periaqueductal gray matter were significantly correlated with pain relief [215]. A brain imaging study (integrated PET/MRI) with the new generation ligand 11C-PBR28 of the translocator protein (TSPO), as a marker of glial activation, demonstrated increased binding to the pain matrix in chronic low back pain patients [216]. The peripheral benzodiazepine receptor, 18 kDa TSPO, is upregulated in activated microglia. PET imaging studies using a specific tracer related to TSPO showed higher activation of the thalamus, anterior and posterior central gyri and paracentral lobule in pain patients versus controls [9].

The inhibition of the activated microglia in NP might be a novel therapeutic target. A CNS glial modulator, propentofylline, administered orally, failed to decrease pain in PHN patients in a proof-of-concept clinical trial (Protocol SLC022/201, EudraCT number 2008-002108-24). Activated p38 MAPK in spinal microglia was detected in peripheral- nerve-injury-associated NP [217]. A RCT of patients suffering from peripheral NP following nerve injury treated with a p38 MAPK inhibitor (dilmapimod) revealed significantly decreased daily average pain scores [217]. An in vitro study revealed that human and rodent microglia responded differently to propentofylline (SLC022) [218]. Another p38 MAPK inhibitor, losmapimod, which was investigated in a RCT of patients with NP from lumbosacral radiculopathy, failed to decrease pain intensity [219].

In a prospective, open-label, pilot RCT of PDN patients, minocycline, a tetracycline antibiotic used as a microglial inhibitor, significantly improved Visual Analog Scale scores [220]. In another clinical trial involving NP patients, minocycline failed to decrease pain intensity [221]. The neuron-microglia interaction is a rate-limiting step in sensitization, both in migraine and NP. The overactivated glial cells have potential effects on the trigeminal ganglia during the pain process in migraine, while they show activation in the dorsal horn of the spinal cord and in the thalamus in NP. However, the results regarding glial biomarkers, like the S100B protein, are controversial in migraine. Modern functional imaging techniques are available for the detection of the presence of hyperactivated glia in the CNS in NP patients. Unfortunately, thus far, microglia inhibitors have failed to reduce pain intensity in these conditions.

## 8. Discussion

In this scoping review, we have highlighted the available clinical data with reference to preclinical studies, starting from CGRP, which has recently gained credit as a target, focusing on the potential pathogenic players both in migraine and NP, and inspiring arguments and the direction of future research.

Hyperexcitability with peripheral and central sensitization leading to allodynia is a common pathological feature in migraine and NP. One of the fundamental players in the pathogenesis of migraine and NP is CGRP. The clinical trials showed that CGRP mAbs reduce pain intensity in both conditions. Several CGRP antagonists and mAbs have been approved for the treatment of migraine, while more clinical studies are expected for CGRP intervention in NP. Regarding the clinical studies of CGRP, many data remain unavailable, including data on interictal M0, MA, CM, and central NP. Hopefully, more clinical studies will reveal the consequences of CGRP modulator administration to clarify its efficacy in those subtypes of pain disorders [222,223,224,225,226]. Different TRP ion channels play distinct roles in both migraine and NP [227]. Early phase clinical trials have begun studying the effect of TRPV-1 receptor antagonists, TRPM-8 receptor agonists, and TRPM-8 receptor antagonists in migraine and studying the effect of TRPV-1 receptor agonists and inhibitors, TRPV-3 receptor antagonists, and TRPM-8 receptor agonists in NP. The clinical data on TRP ion channels are missing for EM and central NP. More clinical research on targeting TRP ion channels is awaited to be know the potential use of TRP receptor modulators for those subtypes. In addition, ATP-sensitive potassium channels are of particular interest for their roles and potential targets in migraine treatment [228].

Migraine and NP share a common pathogenic mechanism with endocannabinoids through the trigeminovascular and pain transmission systems. Potential analgesic lead compounds may be distributed among the cannabinoid system’s receptors and enzymes, such as CBR antagonists and FAAH and MAGL modulators. Clinical studies of peripheral NP on the endocannabinoid remain to be released, and more studies in other subtypes may help ascertain the potential of the system as a target [20,91,92,93]. Another endogenous metabolic system attracting increasing attention for the discovery of analgesic targets is the Trp-KYN metabolic system, which is responsible for the overexcitation of the glutamate nervous system. Monitoring the levels and the ratios of KYN metabolites may be of beneficial use for diagnostic biomarkers and thus, the KYN metabolic system may serve as a potential therapeutic target [229,230,231,232,233,234,235,236,237]. Several clinical studies have reported the status of glutamic acid in different tissue samples taken from migraineurs, but the results remain inconclusive. More studies on KYN metabolites are expected. Clinical studies of central NP remain missing.

Inflammation plays a certain role in both conditions. Preclinical data support the significant involvement of both pro- and anti-inflammatory cytokines [238]. Generally, the levels of proinflammatory cytokines are elevated in migraine and NP, but the participation of anti-inflammatory cytokines and the status of chronic low-grade inflammation in the pathogenesis remain unclear. Relatively more clinical studies were conducted in ictal M0; thus, a systematic review and/or meta-analysis may be able to synthesize the data. More clinical studies on CM and central NP are expected. Clinical data has shown the overactivation of glial function in the pain-relaying neural structures, showing increased S100B and gliosis [150,198]. Thus, the microglia may serve as a potential target by suppressing their activities. Relatively more clinical data on S100B in migraine are available; however, its status remains inconclusive. Clinical studies of glial function are unavailable. More markers for assessing glial function await to be explored.

Overviewing the common potential pathogenic factors in the pathogenesis of migraine and NP in search of novel targets for pain management, the following issues have emerged and may play a role in validating whether the targets we have proposed could remain as main topics of research: the efficacy of CGRP remains uncharted in many subtypes; a better understanding of TRPs in the pathogenesis and research on more TRP modulators remain prerequisites; identifying the key players in the endocannabinoids system and the discovery of more modulators may help research in this field to proceed a step forward; more studies on the status of KYN metabolites in reference to glutamate levels may reveal their potential; a consensus is expected regarding cytokines and sampling to monitor inflammatory status and to zero in on a target; the discovery of more biomarkers and imaging techniques to monitor the status of glial function remains a target (Table 7).

The authors acknowledge the limitations of this review, which include the fact that it has not covered other potential targets and that it has not referred to emerging analgesics that are under extensive research. Matrix metalloproteinases (MMPs) are extracellular matrix metalloproteinases that are implicated in various diseases including migraine and NP. MMPs may play a role in the disintegration of the blood-brain barrier, leading to increased neuronal excitability and thus migraine attacks. Preclinical studies have reported that increased levels of MMPs induce pain-like symptoms, suggesting that MMPs may participate in the pathogenesis of NP and thus could be a potential target for NP. Drug repurposing has helped to identify a cosmetic product as an antimigraine agent. Onabotulinum toxin A (Botox) is a potent neurotoxin that is widely applied in cosmetic procedures. Botox is approved by the FDA for the prophylactic treatment of CM and has been extensively investigated for the potential treatment of NP. The exact mechanisms by which Botox relieves CM and NP remains unknown [239,240]. Furthermore, engineering chimeric compounds targeting more moieties responsible for pain sensation are under extensive study [241,242,243,244,245]. In addition, the contribution of sex and stress hormones such as estrogen and cortisol to pain thresholds and pain sensitivity must be taken into consideration, and may not only help personalized pain management but also elucidate unidentified targets [246].

## 9. Conclusions

This scoping review has recapitulated human data on the pathological components that play a role in the pathogenesis of both migraine and NP with reference to preclinical findings. The data successfully support the hypothesis that migraine and NP have shared pathomechanisms involved in CGRP, TRP ion channels, endocannabinoids, Trp-KYN metabolism, neuroinflammation, and microglial malfunction, clarifying the unchartered areas which are to be explored, reinforcing the need for a better understanding of the mechanisms of those participating components, and inspiring the discovery of more modulators that may provide more options for future research. This line of research, buttressed by preclinical studies, paves an exploratory path forward to help identify new biomarker profiles, develop novel therapeutic agents, and thus build a personalized treatment plan for migraine as well as NP.

## Figures and Tables

**Table 1 ijms-24-04114-t001:** Selected human clinical data related to CGRP in migraine and neuropathic pain.

Migraine					Ref.
EM				CM	
M0		MA			
Ictally	Interictally	Ictally	Interictally		
↑plasma from external jugular vein	-	-	-	-	[41]
↑plasma from cubital vein	-	-	-	-	[42]
Neuropathic pain					ref.
Peripheral NP	Central NP				
↑nerve fibers (painful neuroma)	-				[45]
↑serum (CRPS)	-				[46]
↑keratinocyta (PHN)	-				[47]

↑: increased concentration; ↓: decreased concentration. CGRP: calcitonin gene-related peptide, CM: chronic migraine, CRPS: complex regional pain syndrome, EM: episodic migraine, M0: migraine without aura, MA: migraine with aura, NP: neuropathic pain (no data available), PHN: postherpetic neuralgia.

**Table 2 ijms-24-04114-t002:** Selected human clinical data related to TRP ion channels in migraine and neuropathic pain.

Migraine						Ref.
EM				CM		
M0		MA				
Ictally	Interictally	Ictally	Interictally			
-	-	-	-	↑TRPV-1-like immunoreactive nerve fibers density (the wall of scalp arteries)		[76]
Neuropathic pain						ref.
Peripheral NP					Central NP	
↓pain intensity after 8% capsaicin patch treatment (in post-herpetic neuralgia, chronic postsurgical NP, post-traumatic NP, PDN, HIV-associated NP, painful radiculopathy, trigeminal neuralgia, chemotherapy-induced NP)					-	[20] [77] [78] [79]

↑: increased concentration; ↓: decreased concentration; =: no change. CM: chronic migraine, EM: episodic migraine, HIV: human immunodeficiency virus, IL: interleukin, M0: migraine without aura, MA: migraine with aura, NP: neuropathic pain, nd: no data available, PDN: painful diabetic neuropathy, TRP-1: transient receptor potential 1.

**Table 3 ijms-24-04114-t003:** Selected human clinical data related to endocannabinoids in migraine and neuropathic pain.

Migraine					Ref.
EM				CM	
M0		MA			
Ictally	Interictally	Ictally	Interictally		
-	↑FAAH and AMT (platelet) (only in female patients)	-	-	-	[94]
-	-	-	-	↓anandamide (CSF) ↑PEA (CSF)	[95]
-	-	-	-	↓AMT, FAAH (platelet)	[96]
-	-	-	-	↓anandamide, 2-AG	[97]
-	=anandamide (plasma)	-	=anandamide (plasma)	-	[98]
Neuropathic pain					ref.
Peripheral NP	Central NP				
-	=mean pain intensity after ultramicronized sublingually PEA treatment (NP associated with spinal cord injury)				[99]

↑: increased concentration; ↓: decreased concentration; =: no change. 2-AG: 2-arachidonoylglycerol, AMT: anandamide membrane transporter, CM: chronic migraine, CSF: cerebrospinal fluid, EM: episodic migraine, FAAH: fatty acid amide hydrolase, M0: migraine without aura, MA: migraine with aura, NP: neuropathic pain, -: no data available, PDN: painful diabetic neuropathy, PEA: palmitoylethanolamide.

**Table 4 ijms-24-04114-t004:** Selected human clinical data related to the glutamate and the kynurenine system in migraine and neuropathic pain.

Migraine					Ref.
EM				CM	
M0		MA			
Ictally	Interictally	Ictally	Interictally		
↑glutamic acid (plasma)	↑glutamic acid (plasma)	↑glutamic acid (plasma)	↑glutamic acid (plasma)	-	[118]
-	-	↑glutamic acid (platelet)	↑glutamic acid (platelet)	-	[119]
↓glutamic acid (plasma)	-	↓glutamic (plasma)	-	-	[120]
↑glutamic acid (CSF)	-	↑glutamic acid (CSF)	-	-	[120]
-	↑glutamic acid (saliva)	-	-	-	[121]
-	-	-	-	↑glutamic acid (CSF)	[122]
-	-	-	-	↓KYNA (serum)	[123]
-	↓L-KYN, KYNA, anthranilic acid, picolinic acid, 5-hydroxy-indoleaceticacid (plasma)	-	-	-	[124]
Neuropathic pain					ref.
Peripheral NP				Central NP	
↑L-glutamate (plasma) in CRPS ↓L-Trp (plasma) in CRPS ↑the KYN/TRP ratio				-	[125]
negative correlation: TRP serum level and pain intensity positive correlation: (the KYN/Trp ratio and pain intensity(temporomandibular disorders myalgia)				-	[126]

↑: increased concentration; ↓: decreased concentration. CM: chronic migraine, CRPS: complex regional pain syndrome, CSF: cerebrospinal fluid, EM: episodic migraine, KYNA: kynurenic acid, L-KYN: L-kynurenine, M0: migraine without aura, MA: migraine with aura, NP: neuropathic pain, -: no data available, Trp: tryptophan.

**Table 5 ijms-24-04114-t005:** Selected human clinical data related to cytokines in migraine and neuropathic pain.

Migraine					Ref.
EM				CM	
M0		MA			
Ictally	Interictally	Ictally	Interictally		
↑IL-6 (urine) =IL-1beta (urine) ↓TNF-alpha (urine)	↑IL-6 (urine) =IL-1beta (urine) ↓TNF-alpha (urine)	-	-	-	[158]
=IL-1, TNF (plasma)	=IL-1, TNF (plasma)	=IL-1, TNF (plasma)	=IL-1, TNF (plasma)	-	[159]
-	=TNF-alpha, IL-6 (serum)	-	=TNF-alpha, IL-6 (serum)	-	[160]
-	↑TNF-alpha, IL-1beta, IL-10 (plasma)	-	-	-	[160]
↑TNF-alpha (serum)	-	-	-	-	[161]
↑IL-10, IL-6 (serum)	-	-	-	-	[162]
↑IL-1 receptor antagonist (CSF)	-	↑IL-1 receptor antagonist (CSF)	-	-	[161]
=TN-Falpha (serum)	-	-	-	-	[163]
↑IL-1beta, IL-6 (serum) ↑Il-10 (serum)	-	-	-	-	[164]
↑IL-6 (serum)	-	↑IL-6 (serum)	-	-	[165]
↑TN-Falpha, IL-1beta, IL-6 (serum)	-	-	-	-	[166]
-	↓IL-6 (mRNA and serum)	-	-	-	[167]
-	-	-	-	↑TNF-alpha, IL-6 (serum)	[168]
-	↑IL-1beta (saliva)	-	-	-	[169]
↑IL-18 (serum)	↑IL-18 (serum)	↑IL-18 (serum)	↑IL-18 (serum)	-	[170]
-	-	-	↑IL-6 (mRNA)	-	[171]
-	↑IL-4, IL-18, TGF-beta, TNF-alpha (mRNA) =IL-1beta, IL-17, IL-2 (mRNA)	-	↑IL-4, IL-18, TGF-beta, TNF-alpha (mRNA) =IL-1beta, IL-17, IL-2 (mRNA)	-	[172]
Neuropathic pain					ref.
Peripheral NP				Central NP	
↑TNF-alpha expression (Schwann cells)				-	[173]
↑IL-2, TNF-alpha (mRNA, plasma) ↓IL-4, IL-10 (mRNA, plasma)				-	[174]
↑TNF-alpha (serum) in PDN				-	[175]
↑TNF-alpha (plasma) in PDN				-	[176]
↑IL-6 (serum) in painful DSPN				-	[177]
↑TNF-alpha, IL-1beta (mRNA) ↓IL-10 (mRNA) =IL-4 (mRNA) (in NP after peripheral nerve lesion)				-	[16]

↑: increased concentration; ↓: decreased concentration; =: no change. CM: chronic migraine, CSF: cerebrospinal fluid, DSPN: distal sensori-motor polyneuropathy, EM: episodic migraine, IL: interleukin, M0: migraine without aura, MA: migraine with aura, mRNA: messenger ribonucleic acid, NP: neuropathic pain, -: no data available, PDN: painful diabetic neuropathy, TGF: tumor growth factor, TNF: tumor necrosis factor.

**Table 6 ijms-24-04114-t006:** Selected human clinical data related to glial function in migraine and neuropathic pain.

Migraine					Ref.
EM				CM	
M0		MA			
Ictally	Interictally	Ictally	Interictally		
↑S100B (serum)	↑S100B (serum)	-	-	-	[199]
↑S100B (serum)	↑S100B (serum)	-	-	-	[24]
-	↓S100B (serum)	-	↓S100B (serum)	-	[200]
-	↑S100B (serum)	-	-	↑S100B (serum)	[202]
-	↑S100B (serum)	-	-	↑S100B (serum)	[203]
Neuropathic pain					ref.
Peripheral NP		Central NP			
activated glial cells (PET):thalamus, anterior and posterior central gyrus, paracentral lobule		-			[9,204]

↑: increased concentration; ↓: decreased concentration. CM: chronic migraine, EM: episodic migraine, M0: migraine without aura, MA: migraine with aura, NP: neuropathic pain, -: no data available, PET: positron emission tomography.

**Table 7 ijms-24-04114-t007:** Potential targets for novel pain management and current progress.

Potential Targets	Comments
Calcitonin gene-related peptide (CGRP)	More studies on CGRP modulators in the subtypes are required.
Transient receptor potential (TRP) ion channels	Better understanding of TRPs in the pathogenesis and discovery of more TRP modulators is expected.
Endocannabinoid system	Characterizing the endocannabinoids system network, and the discovery of more enzyme modulators and of the potential use of the metabolites may help advance this field of research.
Tryptophan-kynurenine (KYN) metabolic system	More studies on the status of various KYN metabolites in reference to glutamate levels are expected.
Neuroinflammation	A consensus regarding the type of cytokines and sampling tissues that should be used to monitor an inflammatory status contributing to the pathogenesis should be attained.
Microglia	The discovery of more biomarkers and imaging techniques to monitor the status of glial functions are expected.

## Data Availability

No new data were created or analyzed in this study. Data sharing is not applicable to this article.

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
