# Peer review of "Exploring Novel Therapeutic Targets in the Common Pathogenic Factors in Migraine and Neuropathic Pain"

_ijms, 2023, doi:10.3390/ijms24044114_

Round 1

Reviewer 1 Report

An excellent review. Clearly laid out and comprehensive (although might suggest at least passing mention of MMPs at least in regard to NP).

Minor edits and corrections only. Might also suggest if possible a summary table at end ahead of conclusion - pointing out which mechanisms/mediators show most promise in M0, MA & NP

Author Response

Reviewer 1:

An excellent review. Clearly laid out and comprehensive (although might suggest at least passing mention of MMPs at least in regard to NP).

Response:

We all appreciate your kind comments and valuable suggestions. Also, thank you for your careful language editing. We corrected it accordingly.

Regarding matrix metalloproteinases (MMPs), this review article could not cover all potential new targets and might be discounting several other important elements which are to be highlighted for migraine and NP. We have included MMPs in the discussion section as the limitations as follows:

“The authors acknowledge the limitations of this review which has not covered other potential targets and has not referred to emerging analgesics which are under extensive research. Matrix metalloproteinases (MMPs) are extracellular matrix metalloproteinases, implicated in various diseases including migraine and NP. MMPs may play a role in disintegration of the blood-brain barrier leading to increased neuronal excitability and thus migraine attacks. Preclinical studies reported that increased levels of MMPs induce pain-like behavior, suggesting MMPs may participate in the pathogenesis and thus could be a potential target for NP.”

Minor edits and corrections only. Might also suggest if possible a summary table at end ahead of conclusion - pointing out which mechanisms/mediators show most promise in M0, MA & NP

Response:

Thank you for your careful editing and kind help. We corrected all according the pdf file.

Regarding the tables, we tried to format to reduce the blank part of no date. We have also noticed that the manuscript has succeeded in clarifying currently unavailable data. So, we sincerely hope this review paper also encourage new research to fill the gap in the tables. This is mentioned in the abstract, the introduction, and the conclusion as follows:

Abstract: “Those are several potential analgesic targets which deserve to be explored in search of novel analgesics; however, many evidences remain missing. This review highlights need for more studies on CGRP modifiers for subtypes, discovery of TRP and endocannabinoid modulators, knowledge for the status of KYN metabolite, the consensus on cytokines and sampling, and biomarkers for microglial function, in search of innovative pain management in migraine and NP.”

Introduction: ” Here we highlight clinical studies of common pathogenic factors in migraine and NP with reference to preclinical data to explore potential therapeutic targets and clarify current missing data to be complemented in near future, in search of innovative pain management in migraine and NP.”

Discussion: “Regarding the clinical studies of CGRP, many data remain unavailable, including interictal M0, MA, CM, and central NP. Hopefully, more clinical studies will reveal the consequence of CGRP modulator administration to clarify the efficacy in those subtypes of pain disorders.”

Conclusion: “The data successfully support the hypothesis that migraine and NP have shared pathomechanisms involved in CGRP, TRP ion channels, endocannabinoids, Trp-KYN metabolism, neuroinflammation, and microglial malfunction, clarifying unchartered areas which is to be explored, reinforcing need for better understanding of the mechanisms of those participating components, and inspiring discovery of more modulators which may provide more options for future research.”

In the discussion, an additional table is presented to summarize potential targets and current issued regarding those targets. Hopefully, this table clarify future research direction to reach promising targets:

Table 7. Potential targets for novel pain management and current progress

Potential targets

Comments

Calcitonin gene-related peptide (CGRP)

There need more studies on CGRP modulators in the subtypes.

Transient receptor potential (TRP) ion channels

Better understanding of TRPs in the pathogenesis and discovery of more TRP modulators are expected.

Endocannabinoid system

Characterizing the endocannabinoids system network, discovery of more enzyme modulators and potential us of the metabolites may help advance.

Tryptophan-kynurenine (KYN) metabolic system

More studies on the status of various KYN metabolites in reference to glutamate levels is expected.

Neuroinflammation

A consensus regarding a type of cytokines and sampling tissues to monitor an inflammatory status contributing the pathogenesis should be attained.

Microglia

Discovery of more biomarkers and imaging techniques to monitor the status of glial function are expected.

Reviewer 2 Report

Reviewer comments- IJMS

This manuscript describes “

Exploring Novel Therapeutic Targets in the Common Pathogenic Factors in Migraine and Neuropathic Pain”.  This is an interesting topic of a review article on the common pathogenic factors in migraine and Neuropathic Pain to explore potential novel therapeutic targets. However, there are some minor issues in the current manuscript. This article can be considered for publication.

Minor concerns:

(1)  Introduction is hard to understand without figures. It would be better if authors can add figures to explain pathways and targets.

(2)  Tables are hard to underrated for readers and confusing. So, It needs more clarity. 

(3)  It would better if authors can add or discuss few selected molecules (clinical trial molecules or key molecules on that target or pathways) that targets these pathways. So, readers might find this article more useful. 

(4)  All reference should in uniform pattern.  

Author Response

Reviewer 2:

This manuscript describes “Exploring Novel Therapeutic Targets in the Common Pathogenic Factors in Migraine and Neuropathic Pain”.  This is an interesting topic of a review article on the common pathogenic factors in migraine and Neuropathic Pain to explore potential novel therapeutic targets. However, there are some minor issues in the current manuscript. This article can be considered for publication.

Response:

We all appreciate your kind comments and valuable suggestions. We corrected it accordingly.

Minor concerns:

(1)  Introduction is hard to understand without figures. It would be better if authors can add figures to explain pathways and targets.

Response:

A graphical abstract is added. We hope it helps a reader comprehend the scope of this manuscript.

(2)  Tables are hard to underrated for readers and confusing. So, It needs more clarity. 

Response:

The tables are modified with less white part. No data are replaced with “—” and highlighting missing data has become the purpose of this manuscript, as written in the abstract, the introduction, the discussion, and the conclusion.

Abstract: “Those are several potential analgesic targets which deserve to be explored in search of novel analgesics; however, many evidences remain missing.”

Introduction: ”clarify current missing data to be complemented in near future, in search of innovative pain management in migraine and NP.”

Discussion: “Regarding the clinical studies of CGRP, many data remain unavailable, including interictal M0, MA, CM, and central NP.”

Conclusion: “…, clarifying unchartered areas which is to be explored, ….”

(3)  It would better if authors can add or discuss few selected molecules (clinical trial molecules or key molecules on that target or pathways) that targets these pathways. So, readers might find this article more useful. 

Response:

Some clinical trials are listed where the data are available as follows:

“TRPM-8, a non-selective cation channel, can be activated by cold temperature and menthol. A triple-blind RCT revealed that a 10% menthol solution applied to the forehead and temporal skin areas was significantly superior to the placebo at providing 2-hour pain freedom.”

“The above discussed clinical data led to the design of early phase clinical trials targeting thermoTRP channels for migraine treatment, such as an oral TRPV-1 receptor antagonist (NCT00269022), a TRPM-8 receptor agonist (topical menthol 6%) (NCT01687101) and a TRPM-8 receptor antagonist (oral AMG 333) (NCT01953341) [82].””

Clinical trials revealed that 8% capsaicin patch significantly reduced the average pain intensity in chronic postsurgical NP [88].”

“A clinical trial including patients with painful distal sensorimotor polyneuropathy (DSPN) from the German KORA F4 survey found positive associations between serum concentrations of IL-6 and painful DSPN, whereas no associations were observed for IL-18, TNF-alpha and IL-1 receptor antagonists [194].”

(4)  All reference should in uniform pattern.  

Response:

We corrected it according to the guidelines of IJMS.

Reviewer 3 Report

The authors provide a review of several rather new targets and biomarkers that have been identified in the fight against migraine and neuropathic pain.

The review is generally well written and the authors did collect a good share on relevant literature on the topic. The authors also present the topic with a thematically organized structure, which is critical to produce a clear, memorable and comprehensive view for readers. The provided structure works in terms of splitting the topic in segments of related information, but it is, in my opinion, not ideal in the sense of providing a general framework of understanding. For example, a CGRP related chapter is followed by kynurenine pathway, both providing molecular targets for therapy, then the authors switch to the discussion of glia cell function, represented by a non-therapeutic marker, before discussing the role of cytokines, ion channels and endocannabinoids. A more systematic secondary structure (like f.e. tissue - cell type - pathway - biomarker - therapy) would greatly improve the review. A schematic figure (like in: doi: 10.1111/head.12174, 10.3390/ph12020054) may also help.

Some points that I personally would have expected to be treated were not included in this review. For one, the whole topic of hormonal influence and how it affects therapy has not been treated, which appears to be quite important in this context. Another point missing is the emerging treatment by BoNTA (onabotulinum toxin A). Also, it is not very helpful to mention the migraine generator (line 595) without discussing what it means and the ongoing controversy behind it.

Language may also needs improvement in some parts:

All abbreviations should be explained at first use in text.

line 72: "Microglia are part of neuron-microglia interactions" = pleonasmline 115: "..CGRP released... , lead to.."

line 198: Cav 2.1 (voltage gated ion channel)

line 265: ibudilast is a phosphodiesterase inhibitor. Only specific functions of glia cells can be inhibited.

line 496: "6. Transient receptor potential channels..."

line 554: "Accordingly, they have fundamental..."

Author Response

Reviewer 3:

The authors provide a review of several rather new targets and biomarkers that have been identified in the fight against migraine and neuropathic pain.

The review is generally well written and the authors did collect a good share on relevant literature on the topic. The authors also present the topic with a thematically organized structure, which is critical to produce a clear, memorable and comprehensive view for readers. The provided structure works in terms of splitting the topic in segments of related information, but it is, in my opinion, not ideal in the sense of providing a general framework of understanding. For example, a CGRP related chapter is followed by kynurenine pathway, both providing molecular targets for therapy, then the authors switch to the discussion of glia cell function, represented by a non-therapeutic marker, before discussing the role of cytokines, ion channels and endocannabinoids. A more systematic secondary structure (like f.e. tissue - cell type - pathway - biomarker - therapy) would greatly improve the review. A schematic figure (like in: doi: 10.1111/head.12174, 10.3390/ph12020054) may also help.

Response:

Thank you for your insightful suggestion regarding the secondary structure. We totally agree with the issue. Thus, we have restructured the manuscript in the following order:

CGRP, transient receptor potential ion channel, endocannabinoids, kynurenine metabolism, cytokines, and glial function.

Some points that I personally would have expected to be treated were not included in this review. For one, the whole topic of hormonal influence and how it affects therapy has not been treated, which appears to be quite important in this context. Another point missing is the emerging treatment by BoNTA (onabotulinum toxin A). Also, it is not very helpful to mention the migraine generator (line 595) without discussing what it means and the ongoing controversy behind it.

Response:

We included description on the hormonal and stress factors and onabotulinum toxin A in the discussion section as follows:

“The authors acknowledge the limitations of this review which … and has not referred to emerging analgesics which are under extensive research. … Drug repurposing has identified a cosmetic product as an antimigraine agent. Onabotulinum toxin A (Botox) is a potent neurotoxin widely applied for cosmetic procedures. Botox is approved by the FDA for the prophylactic treatment of chronic migraine and has been extensively investigated for the potential treatment of NP. The exact mechanisms by which Botox relief chronic migraine and NP remains unknown.”

The migraine generator has been rephrased.

Language may also needs improvement in some parts:

Response:

We corrected them as follows:

All abbreviations should be explained at first use in text.

Response:

Thank you. We made a double check.

line 72: "Microglia are part of neuron-microglia interactions" = pleonasmline 115: "..CGRP released... , lead to.."

Response:

“Microglia interact with the neuron.”

“Thus, CGRP released in the peripheral and central branches of trigeminal neurons, lead to vasodilation and neurogenic inflammation of the meninges and the activation of second-order sensory neurons of the TCC.”

line 198: Cav 2.1 (voltage gated ion channel)

Response:

CACNA1A (encoding the α1 subunit of neuronal Ca,2.1. voltage-gated calcium channel)

line 265: ibudilast is a phosphodiesterase inhibitor. Only specific functions of glia cells can be inhibited.

Response:

“a glial cell modulator ibudilast (phosphodiesterase inhibitor) did not reduce the frequency of headache”

line 496: "6. Transient receptor potential channels..."

Response:

“3.Transient receptor potential ion channel function in migraine and neuropathic pain”

line 554: "Accordingly, they have fundamental..."

Response:

“Therefore, they have fundamental roles in nociception and NP transmission”.

Round 2

Reviewer 2 Report

For manuscript "Exploring Novel Therapeutic Targets in the Common Patho-genic Factors in Migraine and Neuropathic Pain", authors have systematically addressed most of concerns  and improved quality of this article. It can be accepted for publication. Good Luck !

Reviewer 3 Report

The manuscript can now be recommended for publication.